# Integrated single-cell functional-proteomic profiling reveals a shift in myofibre specificity in human nemaline myopathy: A proof-of-principle study

Robert A. E. Seaborne[1,2] ID, Roger Moreno-Justicia[3], Jenni Laitila[4,5] ID, Chris T. A. Lewis[2] ID, Lola Savoure[2], Edmar Zanoteli[6], Michael W. Lawlor[7,8], Heinz Jungbluth[9,10], Atul S. Deshmukh[3] ID and Julien Ochala[2] ID

[1]*Centre of Human and Applied Physiological Sciences, School of Basic and Medical Biosciences, Faculty of Life Sciences & Medicine, King's College London, London, UK*

[2]*Department of Biomedical Sciences, Faculty of Health and Medical Sciences, University of Copenhagen, Copenhagen, Denmark*

[3]*Novo Nordisk Foundation Centre for Basic Metabolic Research, Faculty of Health and Medical Sciences, University of Copenhagen, Copenhagen, Denmark*

[4]*The Folkhälsan Research Center, Biomedicum Helsinki, Helsinki, Finland*

[5]*Department of Medical Genetics, Medicum, Biomedicum Helsinki, University of Helsinki, Helsinki, Finland*

[6]*Department of Neurology, Faculdade de Medicina (FMUSP), Universidade de São Paulo, São Paulo, Brazil*

[7]*Department of Pathology, Medical College of Wisconsin, Milwaukee, WI, USA*

[8]*Diverge Translational Science Laboratory, Milwaukee, WI, USA*

[9]*Department of Paediatric Neurology – Neuromuscular Service, Evelina London Children's Hospital, Guy's and St Thomas' Hospitals NHS Foundation Trust, London, UK*

[10]*Randall Centre for Cell and Molecular Biophysics, Faculty of Life Sciences and Medicine (FoLSM), King's College London, London, UK*

Handling Editors: Paul Greenhaff & Kevin Murach

The peer review history is available in the Supporting Information section of this article (https://doi.org/10.1113/JP288363#support-information-section).

**Abstract figure legend** SMPFO uniquely develops data relating to the biophysical state of the sarcomeric protein, alongside global proteome profiling, on a single myofibre basis. Using this approach we found that important myofibre

This article was first published as a preprint. Seaborne RAE, Moreno-Justicia R, Laitila J, Lewis CTA, Savoure L, Zanoteli E, Lawlor MW, Jungbluth H, Deshmukh AS, Ochala J. 2024. Integrated single cell functional-proteomic profiling of human skeletal muscle reveals a shift in cellular specificity in nemaline myopathy. bioRxiv. https://doi.org/10.1101/2024.10.17.618209

The Journal of Physiology

subtype heterogeneity, at both protein function and abundance levels in healthy muscle, is lost or shrunk in myofibres of two subtypes of nemaline myopathy. Created using BioRender.com

**Abstract** Skeletal muscle is a complex syncytial arrangement of an array of cell types and, in the case of muscle-specific cells (myofibres), subtypes. There exists extensive heterogeneity in skeletal muscle functional behaviour and molecular landscape at the cell composition, myofibre subtype and intra-myofibre subtype level. This heterogeneity highlights limitations in currently applied methodological approaches, which has stagnated our understanding of fundamental skeletal muscle biology in both healthy and myopathic contexts. Here we developed a novel approach that combines a fluorescence-based assay for the biophysical examination of the sarcomeric protein, myosin, coupled with same-myofibre high-sensitivity proteome profiling, termed single myofibre protein function-omics (SMPFO). Applying this approach as proof-of-principle we identify the integrated relationship between myofibre functionality and the underlying proteomic landscape that guides divergent, but physiologically important, behaviour in myofibre subtypes in healthy human skeletal muscle. By applying SMPFO to two forms of human nemaline myopathy (*ACTA1* and *TNNT1* mutations), we reveal significant reduction in the divergence of myofibre subtypes across both biophysical and proteomic behaviour. Collectively we demonstrate preliminary findings of SMPFO to support its use to study skeletal muscle with greater specificity, accuracy and resolution than currently applied methods, facilitating that advancement in understanding of skeletal muscle tissue in both healthy and diseased states.

(Received 15 December 2024; accepted after revision 8 April 2025; first published online 28 April 2025)

**Corresponding author** R. Seaborne: Shepherds House, Guy's Campus, King's College London, London, UK. Email: robert.seaborne@kcl.ac.uk; J. Ochala: Department of Biomedical Sciences, Faculty of Health and Medical Sciences, University of Copenhagen, Copenhagen, Denmark. Email: julien.ochala@sund.ku.dk

## Key points

- Skeletal muscle is a complex tissue made up of an array of cell and sub-cell types, with the resident muscle cell – myofibre – critical for contractile function.
- Although single myofibre studies have advanced, existing methods lack the precision for simultaneous multidata analysis, hindering developments in our understanding of skeletal muscle.
- We introduce single myofibre protein function-omics (SMPFO), a method enabling functional analysis of sarcomeric myosin alongside global protein abundance within the same myofibre.
- In healthy myofibres SMyoMFO reveals extensive biochemical diversity in myosin heads, correlating with the abundance of metabolic and sarcomeric proteins, including subtype-specific patterns in sarcoglycan delta (SGCD).
- In contrast SMyoMFO uniquely reveals a reduction in diversity of myosin function and the myofibre proteome in two forms of nemaline myopathy, highlighting disease-associated alterations.
- This innovative approach provides a robust framework for investigating myofibre regulation and dysfunction in skeletal muscle biology.

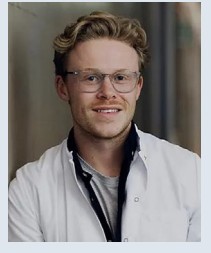

**Robert Seaborne** is a lecturer and principal investigator in muscle biology in the Centre for Human and Applied Physiological Sciences at King's College London. His lab focuses on understanding and deciphering the molecular and functional (dys)regulation of striated muscle at single-cell level. Outside of muscle, Dr. Seaborne has great interest in a range of themes, spanning basic cell biology and functional genomics through to physiological determinants that influence health, disease and exercise performance.

## Introduction

Skeletal muscle (SkM) consists of innervated and metabolically active cells (myofibres) organised into syncytial structures that contain the core contractile apparatus, the sarcomere and the integral myofilaments (e.g. myosin and actin) that facilitate cross-bridge formation and muscular contraction (Frontera & Ochala, 2015). The integrity, function and molecular regulation of SkM are critical for locomotion, respiration and whole-body metabolism. For example the key sarcomeric protein myosin is known to exist in two states of relaxation: the super-relaxed state (SRX) and the disordered-relaxed state (DRX) (McNamara et al., 2015). In the DRX state myosin heads are neither active in functional contraction nor dormant in metabolic activity. Indeed they are actively turnover ATP in orders of magnitude greater than SRX myosin heads, which has significant implications on both tissue and whole-body metabolism (Cooke, 2011). The integrity, functionality and molecular behaviour of SkM and myofibres are increasingly recognised as key determinants of both quality of life and lifespan (Heymsfield, 2024). In genetic SkM disease (myopathies), however, molecular dysregulation impairs SkM function (e.g. dysregulation of SRX/DRX state (Laitila et al., 2024)) and condition, significantly impacting health outcomes in humans. Patients with myopathy, even with similar genetic backgrounds, exhibit considerable heterogeneity in disease onset, symptom manifestation, progression and life expectancy. The basis for such variability remains inadequately understood, but the current lack of knowledge is likely to reflect the inherent complexity of SkM tissue.

Human SkM is a heterogenous composite of cell types, including myofibres, that exist to maintain homeostasis of the tissue. These primary resident SkM cells, myofibres, exist in a spectrum of subtypes ranging from oxidative, fatigue resistant and 'slow' contracting to highly glycolytic and 'fast' contracting fibres (Schiaffino & Reggiani, 2011). The specificity of these discordant myofibres is characterised by a number of important physiological elements, including efficiency, speed and productivity in ATP turnover, motor unit structure and innervation from the motor neuron (Schiaffino & Reggiani, 2011). The precise cellular composition and myofibre subtype proportions found within SkM, and more importantly in the traditionally analysed whole muscle biopsy homogenates, vary significantly depending on internal (e.g. anatomical site) and external (e.g. diseased state) factors, as revealed by recent single-cell analyses (e.g. Kim et al., 2020). Importantly there exist vast differences in the molecular programmes of myofibre and non-myofibre cells, and, as we have recently highlighted, even within myofibre subtype populations (Moreno-Justicia et al.,

2025). These observations highlight that even subtle changes in the underlying composition of (non)myofibre cells within whole muscle homogenates used for analyses will greatly alter the molecular 'snapshot' of the tissue, leading to false positives in datasets (Seaborne & Ochala, 2023).

Recent advances in single-cell approaches have made it possible to overcome cell composition issues within SkM research. Nonetheless traditional single-cell sequencing has two major shortfalls (Seaborne & Ochala, 2023). First as typical single-nuclei approaches use and profile whole sample lysates, they lose the capacity to trace molecular data back to the individual myofibre cell they are derived, thus losing myofibre subtype specificity within the dataset. Although single myofibre sequencing protocols overcome this issue (Blackburn et al., 2019; Sahinyan et al., 2022), they only yield a single dataset per single myofibre, which, given the vast inter-myofibre subtype molecular heterogeneity, falls short at completely profiling single muscle fibres both on molecular and functional levels. Although methods elsewhere have obtained dual-omics data from the same single cell (providing insights into multilayers of molecular regulation; (Angermueller et al., 2016; Argelaguet et al., 2019; Gu et al., 2021)), to date no viable method has been able to develop single-cell paired biophysical and omic data in humans, albeit recent work has been performed in fresh rodent tissue (Ng et al., 2024). In SkM research such an approach would allow improved understanding of how underlying myofibre subtype molecular landscapes may impact functional performance of the cell to a unique level of accuracy and specificity. Combining a functional readout and cell-wide global omics from the same single myofibre will not only help to advance our understanding of SkM biology but also provide a more holistic view of the dysregulation that occurs in SkM disease and in the development of myopathic therapeutics.

Here we present single myofibre protein function-omics (SMPFO), a unique single myofibre workflow, enabling examination of the biophysics of the sarcomeric protein, myosin, coupled with global proteomic profiling. We applied this method to interrogate the myofibre subtype differences in control SkM tissue and in SkM of two forms of human nemaline myopathy, a genetically diverse congenital myopathy with characteristic abnormalities and defects predominantly in proteins affecting normal sarcomeric assembly and contractile function (Jungbluth et al., 2018). Our approach reveals coordinated and physiologically important myofibre subtype specificity at both biophysical and proteome levels, which is largely lost or 'shrunk' in a diseased SkM state. Although performed on low sample number (biopsy and myofibre) and using an uncorrected, compositive *p*-value statistical framework, our work demonstrates the power and applicability of

SMPFO in studying SkM in both healthy and diseased contexts.

## Results

### Single myofibre protein function-omics uniquely correlates myosin biochemical state and global proteome data

We developed a unique workflow, termed SMPFO, to address a gap in the methodological tools available to study single myofibres (Fig. 1*A*). Briefly we first segment single, skinned, resting myofibres into two portions before performing MANT-ATP chase experiments on one segment of the single myofibre. The assay examines the rate of ATP turnover of myosin heads within the myofibre,

with exponential double-decay analyses able to determine percentage of myosin heads in the SRX and DRX state. We couple this assay with global proteomic profiling of the secondary segment of the same original myofibre, providing a molecular snapshot of protein abundance within the same myofibre.

Using this workflow on control SkM biopsies ($N = 8$; Table 1) we successfully resolved both single myofibre global proteome data (Fig. 1*C*) and the conformational state of the key sarcomeric protein myosin, in either the SRX or DRX state (Fig. 1*D*), in 68 of 91 original myofibres analysed ($\sim$75%; Fig. 1*B*). Here in keeping with previous findings from our group (Carrington et al., 2023), we show a highly heterogenous spread of myosin state in healthy, non-contracting/resting SkM, suggesting that myosin exists in an array of poised states for relaxation or contraction (Fig. 1*D*).

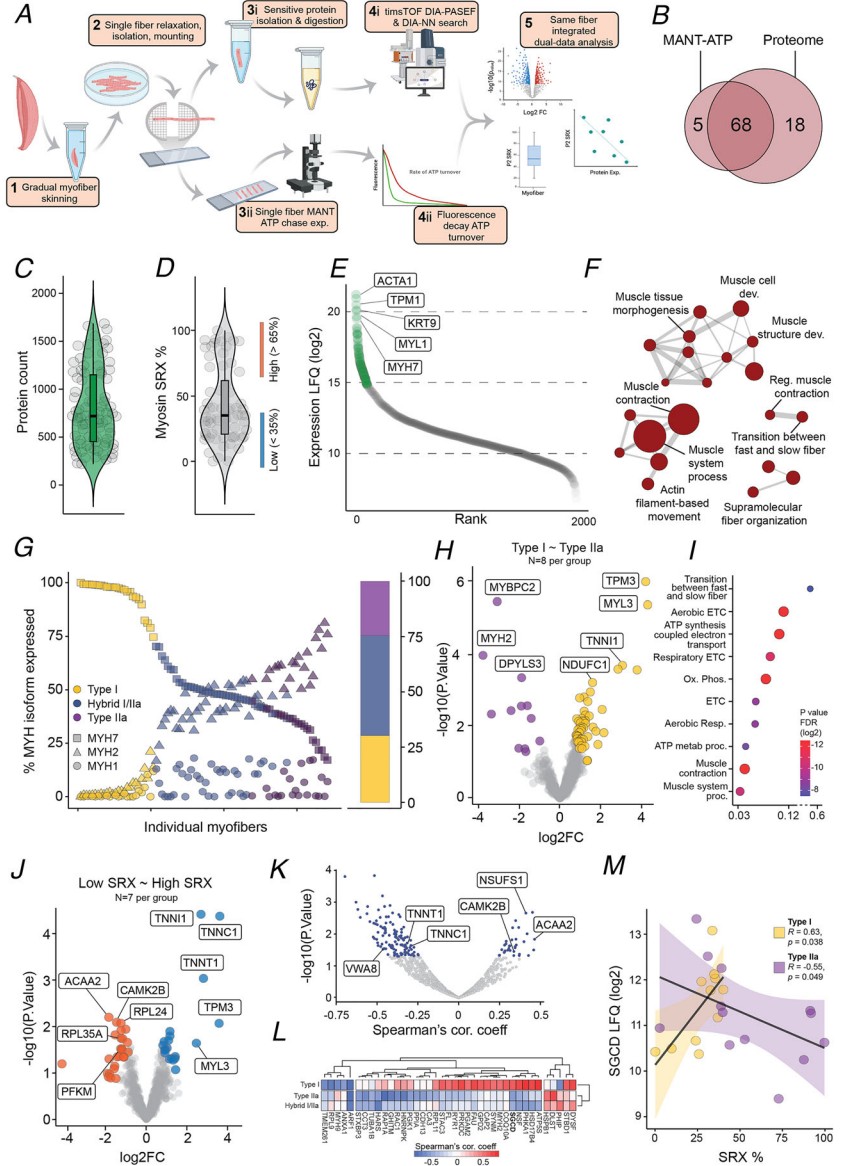

**Figure 1. Single myofibre protein function-omics (SMyoMFO) uniquely integrates single myofibre myosin state with same-cell global proteome profiling**
*A*, schematic representation of the SMyoMFO workflow that enables us to capture both datasets from $N = 68$ single myofibres. *B*, from healthy controls ($N = 8$). *C*, number of quantified protein hits and *D*, state of myosin in SRX% from MANT-ATP chase assay. *E*, single myofibre proteome data hits ranked by expression (LFQ, Log2). *F*, gene ontology (GO) analyses of the 100 most enriched hits mapping to highly specific skeletal muscle terms. *G*, distribution of myofibre subtypes based on the relative expression of MYH7 (square), MYH2 (triangle), MYH1 (dot) within each single myofibre and ranked by the relative expression of MYH7. Inset histogram shows the distribution of type I (yellow), hybrid I/IIa (blue) and IIa (purple) subtypes. *H* and *I*, pseudo-bulk differential abundance analysis of type I (yellow) $\sim$ type IIa (purple) myofibres with significant (unadjusted Xiao threshold $< 0.05$) hits relating to metabolic and muscle GO terms ($N = 8$). *J*, pseudo-bulk analysis performed on single myofibres categorised as either high ($N = 7$, $>65\%$, orange) or low ($N = 7$, $<35\%$, blue) myosin SRX state (denoted in *D*) using unadjusted Xiao significance threshold. *K*, correlation coefficient volcano plot displaying the association between same myofibre SRX% and protein expression across all myofibres (unadjusted *P*-values). On a myofibre subtype level (*L*) we show divergent patterns of association between SRX and protein expression with sarcoglycan delta (*M*) displaying significant divergence between type I and type IIa SRX and protein expression (LFQ log2). Unless otherwise stated $N = 8$ SkM biopsies were used for all analyses.

**Table 1. Descriptive of human muscle biopsy samples**

| Sample | Study | Condition | Sex | Age (years) | Height (cm) | Weight (kg) | Pathogenic variant |
|---|---|---|---|---|---|---|---|
| C1 | Control | Control | M | 57 | 177 | 88.0 | – |
| C2 | Control | Control | M | 60 | 184 | 98.9 | – |
| C3 | Control | Control | M | 61 | 173 | 92.7 | – |
| C4 | Control | Control | M | 62 | 184 | 102.4 | – |
| C5 | Control | Control | M | 49 | 183 | 111.4 | – |
| C6 | Control and disease | Control | M | 23 | 196 | 93.6 | – |
| C7 | Control and disease | Control | M | 22 | 179 | 95.1 | – |
| C8 | Control and disease | Control | M | 30 | 172 | 86.6 | – |
| NM1 | Disease | ACTA1 | M | 3 | – | – | c.611C>T, p.T204I, p.(Thr204Ile) |
| NM2 | Disease | ACTA1 | M | 13 | – | – | c.487C>CG, p.H163D, p.(His163Asp) |
| NM3 | Disease | ACTA1 | F | 11 | – | – | c.158A>T, p.D53V, p.(Asp53Val) |
| NM4 | Disease | TNNT1 | F | 10 (month) | – | – | Homozygous c.661G>T; p.E221X |
| NM5 | Disease | TNNT1 | F | 1 | – | – | Homozygous c.505G>T, p.E180X, p.Glu180Ter |
| NM6 | Disease | TNNT1 | F | 1 | – | – | Homozygous c.505G>T, p.E180X, p.Glu180Ter |

A major benefit of our single myofibre workflow is in its ability to generate single myofibre proteome data, enabling an antibody-independent myofibre subtype classification, beyond classical type I (slow) and type II (fast) isoforms. Using highly sensitive proteome analyses, we successfully obtained an average of 781 ($\pm$401) protein hits from $N = 86$ individual myofibres (Fig. 1*C*; myofibres measuring ~0.5 mm in length), with the most abundant proteins largely specific to key muscle gene ontology (GO) terms such as 'muscle contraction (GO:0006939)', 'muscle cell development (GO:0055001)' and 'muscle system process (GO:0003012)' (Fig. 1*E,F*). Using these data we applied a myosin heavy chain isoform (MYH7, MYH2 and MYH1)-specific expression approach to label each myofibre into specific subtypes, as demonstrated in our recent work (Lewis et al., 2024; Moreno-Justicia et al., 2025), which, in keeping with our previous findings, we resolve type I, hybrid I/IIa and type IIa myofibres, with no evidence of pure type IIx myofibres (Fig. 1*G*). We next grouped type I and type IIa myofibres per sample and performed pseudo-bulk analysis of type I *versus* type IIa myofibres (Fig. 1*H*). Unsurprisingly this comparison revealed the most differentially abundant proteins (differentially abundant proteins; unadjusted Xiao significance threshold < 0.05; Xiao et al., 2014) to belong to both metabolic and sarcomeric GO terms, with 'transition between fast and slow fibre', 'oxidative phosphorylation' and 'muscle contraction' to be some of the most significantly enriched terms (Fig. 1*I*).

Collectively we demonstrate that our SMPFO methodology is able to adequately resolve the biochemical/functional behaviour of the key sarcomeric protein, myosin, as well as developing accurate and specific global proteome data from the exact same single myofibre. To our knowledge this is the first methodology developed to acquire both proteomic and protein functional data from the exact same cell.

## Myosin biochemical state is associated with metabolic and sarcomeric protein abundance in a myofibre subtype-specific manner

The true power of our workflow is in the unique opportunity to pair two separate data types, originating from the exact same myofibre. This enables us, for the very first time, to investigate specific underlying proteome signatures and the association with functional outputs of the myofibre. Using our dual-dataset ($N = 68$; Fig. 1*B*) we first clustered myofibres into either low ($\leq$35%) or high ($\geq$65%) SRX myosin clusters (Fig. 1*D*), before performing pseudo-bulk analyses on these subjects. Interestingly this analysis suggests that myofibres with a low SRX proportion are significantly (unadjusted Xiao threshold of significance < 0.05; Fig. 1*J*) enriched in the abundance of slow isoform sarcomeric proteins (e.g. TNNI1, TNNC1 and TNNT1; Fig. 1*J*). Conversely myofibres with a high SRX percentage have a greater abundance of proteins associated with important roles in metabolic or ATP activity processes (PFKM, CAMK2B, ACAA2) or with roles in ribosomal biology (RPL24, RPL35a). We corroborated these findings by correlating single myofibre SRX with proteomic data from the same myofibre (Fig. 1*K*; unadjusted *p*-value, see methodology). Here by overlapping significant hits identified in both our correlation (Fig. 1*K*) and differential abundance analyses (Fig. 1*J*), we found 18 proteins commonly identified. This list of 18 hits highlights a negative association between SRX proportion and a number of proteins linked with ATPase activity, including VWA8 ($R = -0.46$, $p = 0.029$) and ATP2A2 ($R = -0.29$, $p = 0.045$),

suggesting that greater SRX is associated with a reduction in the abundance of proteins with known ATPase roles.

Taking advantage of our myofibre subtype dataset, we deepened our correlation analysis to a myofibre subtype-specific level (Fig. 1*L*). Analysing only myofibres with both proteome and SRX data, and only proteins that were identified in 50% of myofibres (per subtype analysis, see methods), we identify 39 proteins that significantly correlate in at least one of the myofibre subtypes (Fig. 1*L*). We show clear myofibre subtype specificity within these 39 proteins. For example the protein, sarcoglycan delta (SGCD; Fig. 1*M*), displays significant, but divergent associations between SRX and SGCD abundance in type I ($R = 0.63$, $p = 0.038$) and type IIa myofibres ($R = -0.55$, $p = 0.049$).

Collectively SPMFO highlights the heterogeneity, subtype specificity and relationship of myosin biochemical state and underlying protein abundance in human SkM. These conclusions are drawn based on a low sample (biopsy and myofibre) size and using an unadjusted Xiao threshold of significance, thus requiring independent validation. We have previously shown that SkM myopathies display dysregulation in myosin functionality and the underlying proteomic landscape (Laitila et al., 2024), but the extent of coordination and myofibre subtype specificity in this response is unknown.

## Patients with nemaline myopathy display similarly dysregulated myosin behaviour but diverse myofibre composition

We employed our workflow to SkM biopsies derived from patients with mutations in *ACTA1* ($N = 3$) or *TNNT1* ($N = 3$), as well as controls ($N = 3$; Table 1), from our original analysis. Despite the manually isolated myofibres of *ACTA1* and *TNNT* samples being visibly smaller than control (Fig. 2*A*), our SMPFO method was able to successfully develop a dual-data set of $N = 56$ (80% success rate; Fig. 2*B*). In significant contrast to the array of SRX percentage of myofibres in control SkM myofibres (Con; $54.5 \pm 32.0\%$; Fig. 2*C*), myofibres of *ACTA1*- and *TNNT1*-NM patients are more homogenous, with a shift to a lower SRX proportion (*ACTA1*-, $38.4 \pm 8.6\%$, $P = 0.083$; *TNNT1*-, $33.0 \pm 14.8\%$, $P = 0.012$). This reduction in SRX myosins would suggest an increase in ATP demand and turnover within myofibres of *ACTA1*- and *TNNT1*-SkM (Ranu et al., 2022).

Unsurprisingly the number of identified proteins from our global proteomic analysis was incrementally reduced by disease severity (Con, $1141 \pm 331$; *ACTA1*-, $729 \pm 218$; *TNNT1*-, $635 \pm 256$; Fig. 1*D*), but GO analysis of the most abundantly identified proteins suggested a continued high degree of SkM specificity (Fig. 1*E,F*). Using our myosin heavy chain isoform subtype classification approach, we observed a clear shift in myofibre subtype composition, with *ACTA1*- and *TNNT1*-NM muscle containing a greater percentage of type I and hybrid/type IIa myofibres, respectively (Fig. 2*G*). This observation, supported by immunohistochemistry analyses of control and NM myofibres (Fig. 2*A*), is interesting given that these patients classically present with similar clinical pathology.

## The oxidative metabolic proteome is commonly dysregulated across nemaline myopathy conditions, despite the differing myofibre subtype compositions

We therefore aimed to determine whether the elevated prevalence of specific myofibre subtypes observed in *ACTA1*- and *TNNT1*-associated SkM exhibits comparable or distinct proteomic profiles relative to their respective healthy control types I and IIa myofibres. We performed differential proteomic analysis across specific myofibre subtypes in NM and control samples (Fig. 2*H* and *I*), using unadjusted Xiao threshold of significance to decipher meaningful differences within our data sets. We observed a large disruption in the proteomes of both *ACTA1*-type I (158 differentially abundant proteins) and *TNNT1*-type IIa myofibres (176 differentially abundant proteins) compared to the same control subtypes. When we analysed only the statistically down-regulated proteins in these comparisons, we found surprising commonality in the enrichment of GO terms (Fig. 1*J*). Notably 'aerobic respiration' (GO:0009060) was the most significantly enriched term across both *ACTA1*-type I and *TNNT1*-type IIa analyses, as well as being one of the top 10 most enriched terms in our original control type I ~ type IIa myofibre comparison (Fig. 1*I*). This suggests that, irrespective of present myofibre subtypes, NM induces a ubiquitous dysregulation of the energy and oxidative metabolism-related proteome in NM SkM. Therefore we investigated the extent of this synergistic dysregulation across type I (*ACTA1*-) and type IIa (*TNNT1*-) myofibres in NM *versus* control. We overlapped the differentially abundant proteins from these original differential comparisons (e.g. Fig. 2*H* and *I*), identifying 79 common differentially abundant proteins across NM disease types (Fig. 2*K*). Here an overwhelming proportion of these proteins (78/79) displayed common directionality in dysregulation, including the downregulation of key metabolic proteins such as COX5A, PYGM and PDHA1 (Fig. 2*L*).

## The diversity of the myofibre subtype proteome and the relationship with myosin protein function is lost or 'shrunk' in NM

The only exception to the trend in common directional changes in these 79 proteins was MYBPC2 (Fig 2*L*).

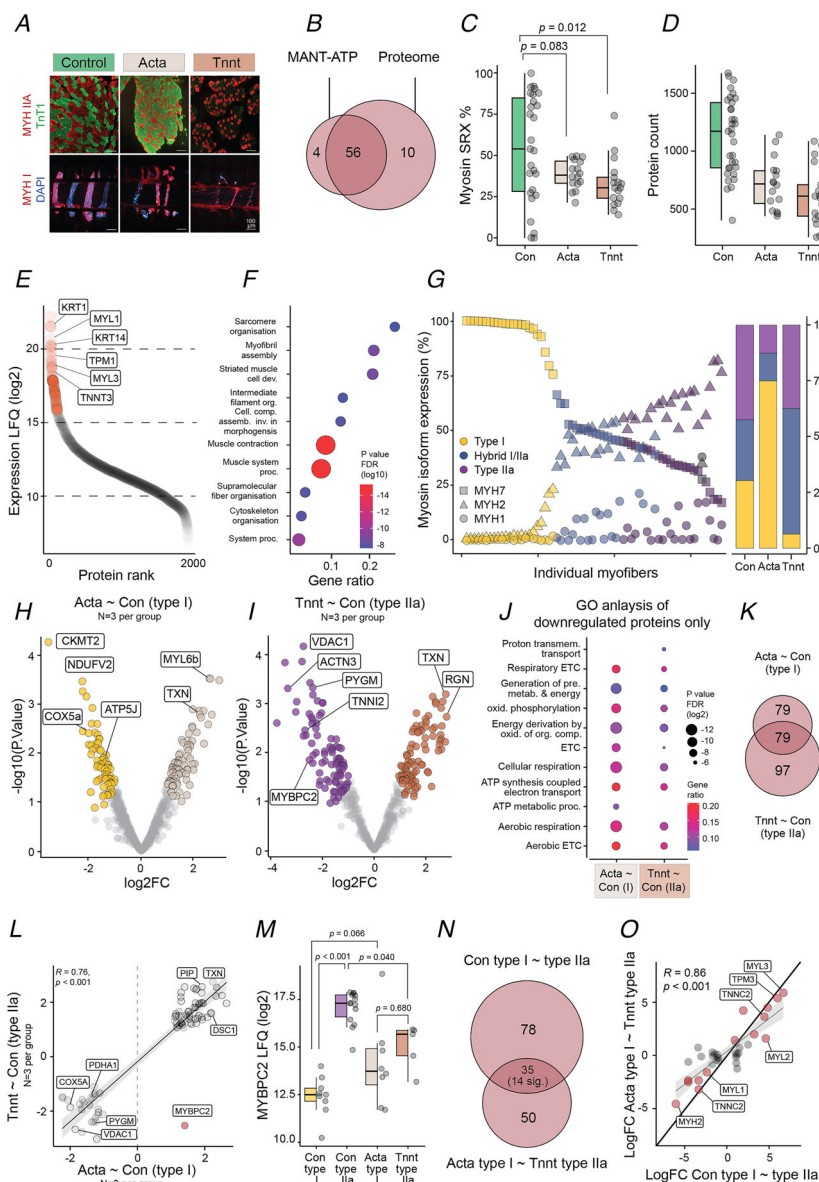

**Figure 2. Single myofibre protein function-omics (SMyoMFO) reveals a shrinking of diversity within both myosin conformational state and protein expression in single myofibres of *ACTA1* and *TNNT1* nemaline myopathy**

*A*, immunofluorescence imaging of skeletal muscle (SkM) cross-sections (upper) and single myofibres (lower) from control, *ACTA1* and *TNNT1* SkM. *B–D*, SMyoMFO captured MANT-ATP (*C*) and global proteomic datasets (*D*) from $N = 56$ single myofibres from control, *ACTA1* and *TNNT1*. *E*, single myofibre proteome data from control, *ACTA1* and *TNNT1* ranked by expression (LFQ, Log2) with gene ontology (*F*) analyses of the 100 most enriched hits suggesting a high specificity for SkM. *G*, as per Fig. 1*G* myofibre subtype distribution and frequency for control, *ACTA1* and *TNNT1* samples, highlighting enriched subtype diversity in patients with nemaline myopathy. *H,I*, differential protein expression analysis (unadjusted Xiao threshold < 0.05) of specific myofibre subtypes, comparing type I (*ACTA1*) and type IIa (*TNNT1*) with relevant controls. *J*, analysis of downregulated proteins from these comparisons reveals commonality in enriched biological terms. *K*, comparative analysis of all differentially expressed proteins highlights distinct overlap in dysregulation between control and *ACTA1*-type I and *TNNT1*-type IIa myofibres. *L*, of the 79 overlapping proteins 78 displayed the same direction of change compared to control. *M*, the exception being MYBC2. In control type I and type IIa myofibres, MYBPC2 expression is significantly different (analysis of variance (ANOVA)). However this differential abundance is reduced in type I and type IIa myofibres of *ACTA1* and *TNNT1*, respectively. $N = 3$/condition, boxplot represents median, interquartile range with single myofibres highlighted by individual points. *N*, representation of differentially expressed proteins (unadjusted Xiao threshold < 0.05) identified in control

type I ∼ type IIa and *ACTA1*-type I ∼ TNNT-type IIa data set. Highlighted are 35 differentially expressed proteins from control analysis that are identified in the entire comparative *ACTA1*-type I ∼ *TNNT1*-type IIa analysis, with only 14 of these proteins retaining significant differential abundance. *O*, unless otherwise stated, $N = 3$ SkM biopsies per condition (control, *ACTA1*, *TNNT1*) for all analyses depicted in the figure.

Indeed MYBPC2 was the only protein significantly dysregulated in both comparisons, but with different directionality of change (Fig. 2*L*). MYBPC2 as the 'fast' paralog of the MYBPC protein family is traditionally more enriched in type II myofibres. Indeed we found MYBPC2 to be one of the most differentially abundant proteins in our control data set (type I ∼ type IIa; Fig. 1*H*) and confirmed this in our disease subset data (Fig. 2*M*). However this myofibre subtype-specific abundance pattern is dysregulated in NM SkM. Indeed MYBPC2 abundance is increased in *ACTA1*-type I myofibres ($14.15 \pm 2.27$) compared to control type I ($p = 0.066$) and significantly decreased in *TNNT1*-type IIa ($15.04 \pm 1.18$) *versus* control type IIa myofibres ($p = 0.040$), resulting in a non-significant difference between the two NM myofibre subtypes (*ACTA1*-type I ∼ *TNNT1*-type IIa, $P = 0.680$; Fig. 2*M*). These data suggest that the myofibre subtype specificity in expression of the fast paralog of the MYBPC family is lost or 'shrunk' in NM.

We sought to ascertain how extensive this 'shrinking' in myofibre subtype diversity in NM SkM is. We theorised that by comparing differentially abundant proteins within our NM myopathy subtype myofibres (e.g. *ACTA1*-type I ∼ *TNNT1*-type IIa) against those found in a control type I *versus* type IIa dataset, we would be able to inspect the degree of maintenance of myofibre subtype diversity across control to NM disease. We performed differential abundance analysis in our matched control (control type I ∼ control type IIa, 78 significant hits, $N = 3$) and NM disease samples (*ACTA1*-type I ∼ *TNNT1*-type IIa, 50 significant hits, $N = 3$), where 35 proteins from our control differentially abundant proteins list were identified in the entire NM dataset (Fig. 2*N*). Of these 35 control differentially abundant proteins, the abundance of 21 proteins was non-significant between *ACTA1*-type I and *TNNT1*-type IIa myofibres (Fig. 2*N*), suggesting a loss of myofibre subtype specificity. A large proportion of these proteins play important roles in mitochondrial function (CKMT2, COA3, COQ9, UQCRC1), energy metabolism/processing (ATPA2, ATP5J, PYGM) and myofibre/sarcomere structure and function in a myofibre subtype manner (ACTN3, MYBPC2, MYH1, MYOM3, MYOZ1, TUBB). The remaining 14 proteins, largely sarcomere isoform-specific proteins, including MYH2, MYL1/2/3, TNNC1/2, TNNT1/2 and TPM3, maintained significance between myofibre subtypes in NM disease (*ACTA1*-type I *vs. TNNT1*-type IIa). However by correlating the differences between myofibre subtypes within each comparison (e.g. control type I ∼ type IIa *vs.*

*ACTA1*-type I ∼ *TNNT1*-type IIa; Fig. 2*O*), we observed a reduction in fold change in NM disease, as highlighted in Fig. 2*O*. This trend is observed across 12 of the 14 significant differentially abundant proteins.

Overall our work, using a novel single myofibre methodology, demonstrates important homeostatic divergence between types I and IIa myofibres in control SkM tissue, which is greatly reduced in the SkM of patients with NM. These conclusions are drawn based on a low sample size and using uncorrected, compositive significance thresholds. These findings now require future validation through independent testing using large sample size and more stringent statistical framework.

## Discussion and summary

Human SkM is a highly heterogenous composition of different cell types and sub-cell types, archetypally observed in the diverse spectrum of myofibre types (Schiaffino & Reggiani, 2011). This complex cellular arrangement is crucial for supporting homeostatic tissue function, but, as we have recently discussed (Seaborne & Ochala, 2023), presents complications when analysing SkM in healthy and diseased contexts. Here to the best of our knowledge we present the first methodology that enables profiling of single-protein biophysical function and global protein expression, originating from the exact same single cell (myofibre) from human SkM. This approach identifies important myofibre subtype coordination between protein abundance and myosin biophysical behaviour in control SkM tissue. Our method shows that such diversity in protein function and abundance is lost, or shrunk, in myofibres of NM patients. Collectively these findings highlight the unique power of SMPFO to transform our understanding of human SkM in both healthy and diseased contexts and to support the advance of therapeutics and interventions to alleviate SkM disorders.

The most important and unique aspect of SMPFO is its ability to resolve functional and protein expression data originating from the same myofibre (termed protein function-omics), allowing highly accurate associations between data sets. Categorising single myofibres into either high or low myosin state, based upon SRX percentage, we identify a signature of protein expression that associates with myosin SRX. The percentage of myofibre SRX is an important variable in SkM tissue and whole-body metabolic demands. Indeed it is purported that if all myofibres switched

from the energy-conservative SRX state to the more ATP-demanding DRX, this would increase whole-body metabolism by ∼50% (Wilson et al., 2021). We show here for the very first time that resting/non-contracting myofibres in a high SRX state have an increased abundance of metabolic-related proteins, including PFKM, CAMK2B and ACAA2, whereas myofibres with a reduced SRX (and concomitant increased DRX) state are enriched for key sarcomeric proteins (TNNI1, TPM3, MYL3). Single myofibre protein function-omic correlative analysis further identifies a negative association between proteins involved in ATP activity (VWA8 and ATP2A2) and the percentage of myosin SRX. Collectively these data suggest that in control SkM tissue myofibres that are in a quiescent, energy-replenished state have an increase in metabolically related proteins, and those in a disordered, energy-consuming state show an increase in proteins that are involved in ATP turnover and sarcomeric organisation.

Single-cell omics technologies have revolutionised our approach to studying SkM, revealing the cell-type diversity, complexity and transcriptomic profiles that control and regulate the tissue (Dos Santos et al., 2020; Kim et al., 2020; Petrany et al., 2020). However due to the underlying chemistry typical single-cell approaches lose the capacity to trace data back to individual myofibre subtypes and cannot yield combined functional and molecular datasets from the same myofibre. By analysing myofibres on an individual basis, SMPFO overcomes these shortfalls, thereby supporting analysis of protein function-omic data on a myofibre-specific subtype basis. Here we correlate myosin SRX percentage with protein abundance in type I and type IIa control myofibres, identifying divergent associations in array of proteins. Interestingly we reveal the correlation between SGCD and SRX percentage that is significantly divergent depending on the specific myofibre subtype analysed. SGCD, a component of the sarcoglycan complex and dystrophic-glycoprotein complex, is involved in linking F-actin cytoskeleton to the extracellular matrix with mutations in the coding gene leading to muscular dystrophy and dilated cardiomyopathy (Coral-Vazquez et al., 1999; Nigro et al., 1996). The myofibre subtype-specific association with SRX percentage is an interesting and unique observation, as very little is known or reported regarding SGCD's potential role in energy consumption in SkM. Thus further work is needed to validate and expand this finding. Collectively we show that SMPFO is able to decipher clear myofibre subtype specificity and diversity across protein function-omic data in control SkM. Such diversity in myofibre behaviour (functional and molecular) is crucial to meet the demands of life, supporting homeostatic regulation of the tissue and promoting health and prolonged lifespan in humans (Bottinelli & Reggiani, 2000).

We therefore employed our method to a diseased SkM context, where there is a dearth in knowledge of molecular pathophysiology, to see if we were able to decrypt some of this obscurity. Herein nemaline myopathy, a subclass of congenital SkM disease, is an archetypal case (Laitila & Wallgren-Pettersson, 2021). In humans mutations in at least 12 causal genes, predominantly implicated in sarcomeric assembly and function, have been identified, leading to a wide spectrum of disease pathology and severity. Histologically it has routinely been observed that patients with pathogenic mutations in actin and troponin present with divergent compositions of myofibre subtypes (enriched for type I and type IIa, respectively), despite presenting with similar disease outcomes (Malfatti & Romero, 2016). Our data are in accordance with these histological observations and also in support of our recent work on the disease (Moreno-Justicia et al., 2025). Interestingly however despite the diversity in myofibre subtype prevalence, we observed relatively similar disruptions in the proteome of these samples. Indeed we report a significant, and ubiquitous, reduction in the abundance of proteins relating to metabolic, oxidative and energy-relating terms that in control SkM underpins a proportion of the diversity between type I and type IIa myofibre subtypes. We also show a clear and significant shrinking of myofibre biophysical behaviour in both forms of NM, irrespective of the underlying myofibre subtype composition. We, and others, have regularly reported that genetic striated muscle disease is associated with a dysregulation in myosin biochemical state (e.g. Anderson et al., 2018; Carrington et al., 2023) and significant disruption of proteins relating to metabolism and energy handling (Laitila et al., 2024; Ranu et al., 2022; Slick et al., 2023; Tinklenberg et al., 2023), but this is the first report illustrating these data originating from the exact same myofibre.

When examining the uniformity between NM types I and IIa proteomes, we identified MYBPC2 abundance to be counterintuitively modified. That is we report an increase in *ACTA1*-type I and reduction in *TNNT1*-type IIa MYBPC2, compared to relevant control myofibres, respectively. MYBPC2 is a key regulator of muscle functionality. Indeed it has been recently reported that homozygous knockout of MYBPC2 induces SkM tissue reductions in grip and plantar flexor muscle strength, as well as speed of contraction and peak isometric force in isolated EDL muscle of mice (Song et al., 2021). Our finding of a reduction in MYBPC2 abundance across specific myofibre subtype in NM is mirrored by a reduction in heterogeneity of myosin SRX state within these same myofibres. There is growing interest in the association between MYBPC and myosin conformation (Lewis & Ochala, 2023). Indeed mice with either homozygous or heterozygous truncating mutations of the cardiac-specific MYBPC paralog (c-MYBPC) report

significantly elevated proportion of myosins in DRX state (Toepfer et al., 2019). Further work shows that not only the genetic state but also the posttranslational state of c-MYBPC (hyperphosphorylated) also induces an increase in myosin DRX state (Jiang et al., 2015). However we are the first to report that a reduction in MYBPC2 differential between type I and IIa myofibres in NM SkM is paralleled by a reduction in myosin relaxed state heterogeneity. This loss of homeostatic myofibre-specific divergence across protein function-omic data, and the impact this has on human SkM disease, requires further examination.

This work is not without limitation. Chiefly our analyses of proteomic data use a non-adjusted, compositive significance threshold of differential findings that incorporates both *p*-value and fold change (Xiao et al., 2014). Although this approach may decipher meaningful patterns within our datasets, it is not sufficiently stringent for affirming biological conclusion. It is therefore worth noting that some findings from our analyses may represent random variance rather than true biological signal. However as a proof-of-principle exploration into the performance and utility of SMPFOs, this manuscript focusses on demonstrating the power and utility of our methodology, rather than establishing firm biological conclusions. Independent validation of these results in future experiments is essential. Second, the sample size of both whole SkM biopsies and the number of myofibres per biopsy, as well as the age and the sex of SkM samples, are divergent across control and myopathy cohorts. These factors limit the experimental and statistical analyses performed and are of vital importance to be addressed in future experiments applying SMPFO to healthy and diseased SkM. The data and findings presented herein must be validated by fresh analyses, using independent samples, greater sample sizes (both muscle biopsy and myofibre number) and more stringent statistical analysis frameworks.

The protocol described here first uses a graded glycerol approach to skin myofibres before performing the dual function-omic assay. Chemical skinning of myofibres is necessary for inspection of myosin biochemical conformation (Stewart et al., 2010) but likely impacts the quantity of proteomic data we were able to retrieve. Although it has recently been suggested that chemical skinning alters structural properties of myofibres (Lewalle et al., 2022), to the best of the authors' knowledge no work has examined the differential proteome content of fresh-skinned myofibres. Nonetheless we report a marked reduction in the number of quantified proteins in these samples compared to our recent work on the same sample set in non-skinned conditions (Moreno-Justicia et al., 2025). This may reflect a reduction in sarcolemma and further membrane-bound proteins (Lewalle et al., 2022). To overcome this postprocessing chemical skinning

by Triton-X100 will enable proteomic analyses to be performed on a non-skinned fraction of the single myofibre. The reduction in quantified targets is also likely due to the portioning of a single myofibre into two constituent parts, each for separate analyses, thus reducing the material for single myofibre proteomic analyses. This latter consideration also implicates sub-myofibre compartmentalisation of molecular landscapes. That is myofibres may possess certain differential molecular domains (e.g. myotendinous junction, Karlsen et al., 2023) along the length of the cell, where our myofibre portioning workflow may impact the myofibre-type specificity of our dual data. Contrary to this hypothesis, however, myofibres seem to possess synchronised and coordinated expression of myosin isoforms (Dos Santos et al., 2020); thus subtype is maintained across the entire myofibre cell. Nonetheless future work would benefit from obtaining all data from the exact same myofibre segment to avoid the confounding of potential molecular compartments within single myofibres.

In summary we have developed the first methodology that allows simultaneous examination of the biophysical functionality of myosin and the global proteome of the exact same single myofibre. This approach provides unique insights into the important myofibre subtype diversity from both a functional and molecular perspective, within control skeletal muscle and the extent to which this is dysregulated, and myofibre subtype diversity shrunk in human nemaline myopathy. SMPFO could therefore become a pivotal tool in not only extending our understanding of fundamental myology and SkM myopathy but also for identifying biomarkers and providing the bases for future therapeutic developments to alleviate SkM disease.

## Methodology

### Sample collection and processing

The samples for this work were obtained from previous studies (outlined where applicable) and are described in Table 1. For our control cohort $N = 8$ patient biopsies were obtained from the *vastus lateralis* muscle under local anaesthesia. Samples were immediately snap frozen in liquid nitrogen and stored for further processing (Sahl et al., 2018; Søndergård et al., 2021). These samples were approved by the ethical committee for the Capital Region of Denmark (H-15010122) and by the local ethics committee of Copenhagen and Frederiksberg (H-15002266) with full participant consent. Six patients with confirmed cases of nemaline myopathy ($N = 3$ *ACTA1-*, $N = 3$ *TNNT1-*) were selected for analysis in our disease study, alongside $N = 3$ control samples from our original control cohort (Table 1). The biopsies from patients with nemaline myopathy were collected,

stored and processed in accordance with the Human Tissue Act under local ethical in United Kingdom (REC 13/NE/0373) with consent. All procedures were carried out in accordance with the Declaration of Helsinki. From all samples a section of the original biopsy was dissected and isolated under sterile, frozen conditions and prepared for processing.

### Solutions and buffers

For Mant-ATP chase experiments relaxing solution contained 4 mM Mg-ATP, 1 mM free $Mg^{2+}$, 10–6 mM free $Ca^{2+}$, 20 mM imidazole, 7 mM EGTA, 14.5 mM creatine phosphate and KCl to an ionic strength of 180 mM and pH of 7.0. Rigor buffer contained 120 mM K acetate, 5 mM Mg acetate, 2.5 mM $K_2HPO_4$, 50 mM MPS and 2 mM DTT with a pH of 6.8.

For proteomics lysis buffer contained 1% sodium dodecyl sulphate, 40 mM chloroacetamide, 10 mM dithiothreitol in 50 mM tris with a pH of 8.5.

### Equations

Double exponential Mant-ATP decay:

$$\text{Normalised fluorescence} = 1 - P_1 \left(1 - \exp^{(-t/T_1)}\right)$$
$$- P_2 \left(1 - \exp^{(-t/T_2)}\right)$$

Note: $P_1$ = amplitude of the initial decay; $T_1$ = time constant for $P_1$; $P_2$ = amplitude of the secondary decay; $T_2$ = time constant for $P_2$.

### Single-fibre dual-assay processing

Individual myofibres were then transferred to microscopy slides containing single half-split copper meshes designed for electron microscopy (G100 2010C:XA; SPI Supplies, West Chester, PA, USA; width, 3 mm). Myofibres were clamped under copper grids with N of 5/6 per copper mesh per microscopy slide. Once clamped the end tail of each individual myofibre was dissected, and this micro-piece of myofibre was transferred via micro-tweezers to a single well of a 96-well plate on ice, containing proteomic lysis buffer (15 ul; see above). Between the handling and dissection of each individual myofibre, implements (micro-tweezers, scalpel, syringe needles) were washed through with RNAse zap, 100% EtOH and left to air-dry. Multiple sets of implements were used during experiments to ensure efficient and processing of all myofibres. Herein each single myofibre pertained two micro-fragments, one piece for MANT-ATP chase experiments and one piece for single myofibre global proteomic profiling (Fig. 1A).

### MANT-ATP chase assay

Non-contracting/resting single myofibres were mounted in accordance with our previous studies (Laitila et al., 2024; Lewis et al., 2024; Ochala et al., 2021). Cover slips were placed on the top of the microscopy slide, held with double-sided tape to generate 'flow' chambers where myofibres were incubated in rigor buffer for 5 min at 25°C. A rigor buffer containing 250 um Mant-ATP was then flushed through the chamber and left to incubate the myofibres for 5 min. A secondary rigor buffer containing unlabelled ATP (4 Mm) was flushed into the chamber, and acquisition of the Mant-ATP chase occurred.

### MANT-ATP fluorescence acquisition

To acquire fluorescence decay in individual myofibres a microscopy set-up containing AxioCam ICm1 camera (Zeiss) with Plan-Apochromat 20X/0.8 objective and Axio Scope A1 microscope (Zeiss), took images (20 ms exposure time using a DAPI filter set) every 5 s for the first 90 s and every 10 s for the remaining 5 min acquisition time. From each myofibre three independent regions of the same myofibre were sampled for fluorescence decay using region of interest (ROI) manager (ImageJ; Bethesda, MD, USA). A normalised mean fluorescence intensity value was generated by taking the background intensity away from each individual image, averaged across the three images and normalised to the final Mant-ATP image ($T = 0$). Data were exported to Prism (V9.0; GraphPad Software Inc., San Diego, CA, USA) and fitted to an unconstrained double exponential decay (see equations). The double exponential decay indicates the initial rapid decay (P1) as percentage of myosin in DRX state, and the slower, secondary decay (P2) as the percentage of myosin in the SRX state (Ochala et al., 2021).

### Myofibre proteomic sample preparation

Single muscle fibre proteomics was performed as recently described. In brief myofibre fragments were placed into 10–15 µL of proteomic lysis buffer (1% sodium dodecyl sulphate, 40 mM chloroacetamide, 10 mM dithiothreitol in 50 mM tris, pH 8.5) and boiled at 95°C in a thermomixer (800 rpm gentle shaking) before sonication with a bioruptor (30 s on/off cycle totalling 15 min). Myofibre lysates were brought up to 50 µL with remaining lysis buffer and underwent protein digestion by the addition of both trypsin (1:100; Promega) and LysC (1:500; Wako) in respective enzyme to protein ratios. The reaction was allowed to take place overnight in a thermomixer set at 37°C and 800 rpm. The next day protein digestion was quenched by addition of 2% trifluoroacetic acid in isopropanol, and the peptides were desalted using in-house prepared

single-use reverse-phase StageTips containing styrene divinylbenzene reverse-phase sulfonate (SDB-RPS) discs. Lastly desalted peptides were loaded into Evotips (Evosep, Odense, Denmark) following manufacturer's loading protocols prior to liquid chromatography tandem mass spectrometry (LC–MS/MS) analysis.

### Liquid chromatography tandem mass spectrometry

The LC–MS/MS instrumentation consisted of an Evosep One HPLC system (Evosep, Odense, Denmark) (Bache et al., 2018) coupled via electrospray ionisation to a timsTOF SCP mass spectrometer (Bruker, MA, America). Peptides were separated using an 8 cm, 150 μM ID column with C18 beads (1.5 μm). Chromatographic separation followed the '60 samples per day' method, and electro-spray ionisation was performed via a CaptiveSpray ion source and a 10 μm emitter into the MS instrument. Single muscle fibre peptides were measured using DIA-PASEF (Meier et al., 2020), with a scan range of 400–1000 $m/z$. The TIMS mobility range was set to 0.64–1.37, and cycle time was 0.95 s, using 8 DIA-PASEF scans.

### Data processing

Raw MS spectra were processed using the DIA-NN software (version 1.8) (Demichev et al., 2020). The search was conducted in a library-based manner, using a previously developed myofibre-specific library (Moreno-Justicia et al., 2025). Proteotypic peptides were selected for quantification, and the neural network was operated in double-pass mode. Robust LC (high accuracy) was the quantification strategy of choice, together with the match between runs option and a precursor false discovery rate (FDR) of 1%. Unless specified other DIA-NN settings remained as default. Downstream bioinformatics were conducted using the PG_matrix file from the DIA-NN output. Principal component analysis was performed using Log2-transformed LFQ values for all myofibres, using *prcomp()* omitting NA values. This revealed one control myofibre (sample; C7, type; I/IIa hybrid) to be an outlier that was removed. Fibre typing of individual myofibres for both control and NM cohorts was performed as previously described (Moreno-Justicia et al., 2025) with small modifications. Briefly on a single myofibre basis, the raw LFQ values for three myosin isoforms (MYH1, MYH2 and MYH7) were used to calculate their respective relative abundance. Samples were then ordered by MYH7 from high to low, and an absolute threshold in abundance was used to assign each myofibre to a specific subtype (type I, type IIa, hybrid I/IIa).

### Differential abundance

Pseudo-bulk differential abundance analysis was performed by mathematically downsampling each participant's total data to create one median data value per protein hit per MYH-based fibre type. The limma (version 3.54.2) workflow was followed, including a quantile normalisation of all samples and differential abundance (linear model) analysis across fibre sub-types in control cohort and across conditions in the NM cohort. The Xiao significance score was applied to identify significant proteins (<0.05), which considers the fold change and statistical significance of protein hits (Xiao et al., 2014). Although this approach does not provide definitive biological conclusions to be drawn, it allows for meaningful patterns to be deciphered from our datasets. To note for SRX% differential proteome analysis, $N = 7$ samples contained both low (≤35%) and high (≥65%) SRX myofibre subtypes and were used for analysis.

### Functional enrichment

GO functional enrichment analysis was performed focusing on 'biological process' terminologies, unless otherwise stated. List of relevant protein hits (identified within relevant analyses) was exported to ShinyGO (version 0.77 or version 080; Ge et al., 2020), with the entire dataset or comparative dataset used as associated background, to ascertain functional enrichment terms with an adjusted FDR cut-off of <0.05 interpreted as significant. Gene ratio was ascertained by dividing the number of identified genes in the input list by the total number of genes within the term. Where relevant, packages REVIGO (version 1.8.1 (Supek et al., 2011)), MetaScape (Zhou et al., 2019) and Cytoscape (version 3.9.1) were used to develop and represent GO association networks.

### Correlation coefficient and hierarchical clustering

Data were filtered to remove outlier ($N = 1$) and non-dual data ($N = 23$) samples and protein data present in <22 of these samples, retaining a dataset of $N = 67$ samples and $N = 979$ protein hits. Log2 LFQ protein values were correlated (Spearman, unadjusted) against SRX and DRX% in remaining samples via *rcorr()* function in Hmisc package (version 5.1-3). Myofibre sub-type correlation analysis was performed as identical to that above, with the exception that protein hits must be present in ≥10 samples of each myofibre subtype. Hierarchical clustering was performed on significant type I and type

IIa $\sim$ SRX correlations, using *pheatmap()* function from the Pheatmap package (version 1.0.12). Clusters were determined by visual inspection performed via *cuttree()* function.

### Immunolabelling of muscle sections

Cryosections (10 um) were fixed in 4% PFA (10 min), permeabilised with Triton X-100 (0.1% for 20 min) and blocked in 10% normal goat serum (500627, Life Technologies) supplemented with 0.1% BSA (1 h). Sections were incubated overnight with diluted (1:25) primary antibodies against MYH7 (mouse monoclonal A4.951, Santra Cruz, sc-53090) or MYH2 (mouse monoclonal SC71, DSHB) combined with a further antibody against TNNT1 (1:500; rabbit polyclonal HPA058448, Sigma) in 5% goat serum (supplemented with 0.1% of BSA and Triton X-100). Alexa Fluor Goat anti-Mouse 647 (A21237) and Alexa Fluor Dinkey anti-Rabbit 448 (A11034) were used for primary MYH and TNNT1 antibodies, respectively (1:500 dilution in 10% normal goat serum). A fluorescence microscopy set-up (10× objective, Zeiss Axio Observer 4 fluorescent microscope, Colibri 5 led detector, Zeiss Axiocam 705 mono camera, Zen Software, Zeiss) captured images of all conditions. A selection of myofibres were mounted on copper grids and imaged through a stereomicroscope for visualisation purposes. Fluorescent images were obtained with a 10× objective on a Zeiss Axio Observer 3 fluorescence microscope with a Colibri 5 led detector, combined with a Zeiss Axiocam 705 mono camera, using Zen software (Zeiss). For visualisation purposes, a selection of fibres were mounted on copper grids glued on a microscopy slide and imaged under a stereomicroscope.

### Statistical analysis

All data, unless otherwise stated, are presented as means $\pm$ standard deviation (SD) and where possible presented with individual data points. For all Mant-ATP experiments, data were prepared in Prism (V9.0), exported to R Studio via .csv format where it was statistically analysed (ANOVA, with Tukey's honest significant difference (HSD) *post hoc*) to test for significant ($p < 0.05$) differences between control ($N = 27$), *ACTA1* ($N = 16$), *TNNT1* ($N = 17$). Log2-transformed LFQ values for individual protein targets were extracted and an ANOVA with Tukey's *post hoc* used to compare across groups/conditions (RStudio). Statistical analysis of proteomic data is detailed in relevant subsections.

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

## Additional information

### Data availability statement

All raw proteomics data have been deposited on the ProteomeXchange Consortium via the PRIDE partner repository (Perez-Riverol et al., 2025) under the accession number PXD061287. The code used to analyse proteomic datasets is available at https://github.com/RobertSeaborne/SMF-protein-function-omics.git.

### Competing interests

C.T.A.L. is an employee at Novo Nordisk A/S. Their contribution to this study was carried out prior to this employment and has no influence on the results presented or conclusions drawn in this study. This manuscript was originally submitted as a preprint to the bioRxiv preprint server for biology (Seaborne et al., 2024).

### Author contributions

Conceptualisation; R.A.E.S., A.S.D., J.O. Data curation; R.A.E.S., R.M.-J., J.L., C.T.A.L., L.S., A.S.D., J.O. Formal analysis; R.A.E.S., R.M.-J., J.L., C.T.A.L., A.S.D., J.O. Funding acquisition; R.A.E.S., A.S.D., J.O. Investigation; R.A.E.S., R.M.-J., J.L., C.T.A.L., L.S., E.Z., M.W.L., A.S.D., J.O. Methodology; R.A.E.S., R.M.-J, A.S.D., J.O. Project administration; R.A.E.S., J.O. Visualisation; R.A.E.S. Writing – original draft; R.A.E.S., J.O. Writing – review

& editing; R.A.E.S., R.M.-J., J.L., C.T.A.L., L.S., E.Z., M.W.L., H.J., A.S.D., J.O.

## Funding

R.A.E.S was funded by a Lundbeck Postdoctoral Fellowship (R347-2020-654) and a Talent Prize research award (R417-2022-1294). This work was generously funded by the Novo Nordisk Foundation (NNF21OC0070539) and Lundbeckfonden (R434-2023-311) to J.O., as well as an unconditional donation from the NNF to NNF Centre for Basic Metabolic Research (grant numbers NNF18CC0034900 and NNF23SA0084103). Mass spectrometry analyses were performed by the Proteomics Research Infrastructure (PRI) at the University of Copenhagen (UCPH), supported by the NNF (grant agreement number NNF19SA0059305).

## Acknowledgements

We thank Thomas Nyegaard Beck for his assistance in many of the experiments outlined in the present manuscript. Mass spectrometry was performed by the Proteomics Research Infrastructure (PRI) at the University of Copenhagen (UCPH), supported by the Novo Nordisk Foundation (NNF19SA0059305).

## Keywords

human muscle, myopathy, myosin, proteomics

## Supporting information

Additional supporting information can be found online in the Supporting Information section at the end of the HTML view of the article. Supporting information files available:

**Peer Review History**

