## [Peer Review History · The Journal of Physiology]

Integrated single cell functional-proteomic profiling reveals a shift in myofiber specificity in human nemaline myopathy: a proof-of-principle study

Robert A Seaborne, Roger Moreno-Justicia, Jenni Iaitila, Christopher T.A. Lewis, Lola Savoure, Edmar Zanoteli, Michael Lawlor, Heinz Jungbluth, Atul S Deshmukh, and Julien Ochala

DOI: 10.1113/JP288363

Corresponding author(s): Robert Seaborne (robert.seaborne@kcl.ac.uk)

Review Timeline:

Submission Date:	15-Dec-2024
Editorial Decision:	07-Jan-2025
Revision Received:	05-Mar-2025
Editorial Decision:	25-Mar-2025
Revision Received:	02-Apr-2025
Accepted:	08-Apr-2025

Senior Editor: Paul Greenhaff

Reviewing Editor: Kevin Murach

Transaction Report:

Dear Dr Seaborne,

Re: JP-RP-2024-288363 "Integrated single cell functional-proteomic profiling of human skeletal muscle reveals a shift in cellular specificity in nemaline myopathy" by Robert A Seaborne, Roger Moreno-Justicia, Jenni Iaitila, Christopher T.A. Lewis, Lola Savoure, Edmar Zanoteli, Michael Lawlor, Heinz Jungbluth, Atul S Deshmukh, and Julien Ochala

Thank you for submitting your manuscript to The Journal of Physiology. It has been assessed by a Reviewing Editor and by 2 expert referees and we are pleased to tell you that it is potentially acceptable for publication following satisfactory major revision.

REVISION CHECKLIST:

We look forward to receiving your revised submission.

Yours sincerely,

Paul Greenhaff
Senior Editor
The Journal of Physiology

REQUIRED ITEMS

- Include a Key Points list in the article itself, before the Abstract.
- Author photo and profile. First or joint first authors are asked to provide a short biography (no more than 100 words for one author or 150 words in total for joint first authors) and a portrait photograph. These should be uploaded and clearly labelled together in a Word document with the revised version of the manuscript. See Information for Authors for further details.
- Please ensure that any tables are editable and in Word format, and wherever possible, embedded in the article file itself.
- Please ensure that the Article File you upload is a Word file.
- Your paper contains Supporting Information of a type that we no longer publish, including supplementary tables and figures. Any information essential to an understanding of the paper must be included as part of the main manuscript and figures. The only Supporting Information that we publish are video and audio, 3D structures, program codes and large data files. Your revised paper will be returned to you if it does not adhere to our Supporting Information Guidelines.
- Papers must comply with the Statistics Policy: https://jp.msubmit.net/cgi-bin/main.plex?form_type=display_requirements#statistics.

In summary:

- If $n \leq 30$, all data points must be plotted in the figure in a way that reveals their range and distribution. A bar graph with data points overlaid, a box and whisker plot or a violin plot (preferably with data points included) are acceptable formats.
- If $n > 30$, then the entire raw dataset must be made available either as supporting information, or hosted on a not-for-profit repository, e.g. FigShare, with access details provided in the manuscript.
- 'n' clearly defined (e.g. x cells from y slices in z animals) in the Methods. Authors should be mindful of pseudoreplication.
- All relevant 'n' values must be clearly stated in the main text, figures and tables.
- The most appropriate summary statistic (e.g. mean or median and standard deviation) must be used. Standard Error of the Mean (SEM) alone is not permitted.
- Exact p values must be stated. Authors must not use 'greater than' or 'less than'. Exact p values must be stated to three significant figures even when 'no statistical significance' is claimed.
- Please include an Abstract Figure file, as well as the Figure Legend text within the main article file. The Abstract Figure is a piece of artwork designed to give readers an immediate understanding of the research and should summarise the main conclusions. If possible, the image should be easily 'readable' from left to right or top to bottom. It should show the physiological relevance of the manuscript so readers can assess the importance and content of its findings. Abstract Figures

should not merely recapitulate other figures in the manuscript. Please try to keep the diagram as simple as possible and without superfluous information that may distract from the main conclusion(s). Abstract Figures must be provided by authors no later than the revised manuscript stage and should be uploaded as a separate file during online submission labelled as File Type 'Abstract Figure'. Please also ensure that you include the figure legend in the main article file. All Abstract Figures should be created using BioRender. Authors should use The Journal's premium BioRender account to export high-resolution images. Details on how to use and access the premium account are included as part of this email.

- Please ensure that all figures and tables have a title and legend, and that they have been cited within the main article text. Figure 2 is not currently cited.

EDITOR COMMENTS

Reviewing Editor:

You work has been evaluated by two experts in the field. Both agree that this work is impactful and will provide a valuable resource to the field, in addition to presenting some interesting findings. There are concerns regarding the statistics, some missing details in the Methods, and lack of data transparency that must be addressed. Overall, though, this work is viewed favorably and can be considered further after revision.

There were concerns from Reviewer #1 regarding the reporting of the statistics, and exact p values are missing. Code for analysis as well as raw data is also missing and should be provided in a public format.

Senior Editor:

Comments to the Author:

Thank you for the manuscript submission to The Journal of Physiology. It has been considered by a reviewing editor and two expert reviewers, and I am pleased to say the general consensus is favourable. Nevertheless, both reviewers have raised points for the authors to consider, including a number of significant concerns by Reviewer 1. It is also important that the authors pay close attention to these points and the instructions for authors when revising the research article so as to meet TJP guidelines: https://jp.msubmit.net/cgi-bin/main.plex?form_type=display_requirements#Revised%20submissions

REFEREE COMMENTS

Referee #1:

This manuscript appears that it may produce important results but there are some major flaws that make it unclear and difficult to review as follows:

1. The paper should include some description about what the functional assay is measuring and what that means functionally. Right now that is assumed. For example, SRX is not defined anywhere. What are the limitations of this functional measurement? Are there known biases of this approach for each specific fiber type?
2. Figure legends are missing making figures harder to interpret
3. More detail on the proteomic sample processing should be given in the manuscript instead of relying on reference to a previous paper.
4. Figure 1G and 2G are unclear and there seems likely a mistake in their preparation. The y-axis is the % of which MYH isoform? It seems this figure would require at least two panels showing the % of MYH7 and a second showing the percent of MYH2?
5. Figure 1M - the points should be colored according to which group they are in to enhance clarity.
6. There appears to be no mention of multiple testing correction anywhere, which is required any time multiple tests are performed. Please use corrected p-values for all comparisons. For example, use Benjamini Hochberg for 1H, 1J, 1K, 1L, 1M, 2H, and 2I.
7. Despite major differences in proteomic depth between control and myopathy, the data appears to have been beautifully normalized (Fig 2H and 2I). The statistical section of the methods is vague with regard to how this was achieved. Please include the code for all data analyses on github or zenodo and include more details in the methods section.
8. All the raw mass spectrometry data must be shared publicly and checked before manuscript acceptance. That section of the method currently lacks the data coordinates.
9. Why were type 1 fibers compared to control only for ACTA1- and why were type 2a only compared to control for TNNT-

fibers?

10. What is $Xiao < 0.05$?

Referee #2:

This manuscript reports the development and use of a two-step approach to describe single myofiber proteomic characterization coupled with a myosin biophysical ATPase measurement. The novel approach is reported for fibers from health humans and is then advanced to two muscle diseases. A rich dataset here and in future studies using this novel approach will be valuable to the scientific community. Additions noted below may introduce possibilities to further advance the field.

1) Introduction:

- a. Please use patient-first (person-first) language, e.g., rather than "Myopathy patients" write "Patients with myopathy".
- b. There is a lack of information on myosin SRX-DRX states which is needed for readers not familiar with resting myosin ATPases; a couple of sentences on this is warranted earlier than that in the results and discussion.
- c. Along with the brief description of myosin SRX, a sentence rationalizing the prioritization of this resting (non-contraction) sarcomeric characterization would be helpful as well, again prior to what is written in the discussion.

2) Results:

- a. Avoid the word "shape" as this implies that protein conformation or an imaging technique was utilized to evaluate myosin.
- b. Can any deduction be made in regard to thick vs thin filament proteins, for example, within the results for the nemaline myopathy fiber measurements? Or, in Fig 1 J, thin filament proteins tend to cluster (top blue) apart from thick filament proteins (bottom blue). Is this trend more noticeable than these 5 proteins labeled?
- c. In Fig 1A and elsewhere, it should be made clear that the ATPase measured is during relaxation/ in a resting fiber.
- d. Fig legends for Fig 1 and 2 are missing from the composite pdf.

3) Discussion:

- a. In describing advanced characteristics of 'fast' and 'slow' fibers, it seems that the role of the motor neuron should minimally be noted.
- b. Another limitation not mentioned is sex and age discrepancies of patients in the control vs myopathy groups, i.e., controls are all male while 4 of 6 in the myopathy group are female. How might the very large age differences in control vs myopathy groups affect the results?

END OF COMMENTS

Author response to address editorial and reviewer comments, with regards:

Manuscript: Integrated single cell functional-proteomic profiling of human skeletal muscle reveals a shift in cellular specificity in nemaline myopathy

EDITOR COMMENTS

Reviewing Editor:

Your work has been evaluated by two experts in the field. Both agree that this work is impactful and will provide a valuable resource to the field, in addition to presenting some interesting findings. There are concerns regarding the statistics, some missing details in the Methods, and lack of data transparency that must be addressed. Overall, though, this work is viewed favourably and can be considered further after revision.

There were concerns from Reviewer #1 regarding the reporting of the statistics, and exact p values are missing. Code for analysis as well as raw data is also missing and should be provided in a public format.

Senior Editor:

Comments to the Author:

Thank you for the manuscript submission to The Journal of Physiology. It has been considered by a reviewing editor and two expert reviewers, and I am pleased to say the general consensus is favourable. Nevertheless, both reviewers have raised points for the authors to consider, including a number of significant concerns by Reviewer 1. It is also important that the authors pay close attention to these points and the instructions for authors when revising the research article so as to meet TJP guidelines: https://jp.msubmit.net/cgi-bin/main.plex?form_type=display_requirements#Revised%20submissions

Author Response: To the reviewing and senior editor. Thank you for the efficient processing of our manuscript and timely evaluation. We have addressed the outstanding points with regards manuscript guidelines for submission to J. Phys. Below, we have responded in point-by-point fashion to the comments from reviewers 1 and 2, with our responses highlighted in red text. If/where appropriate several comments by the single reviewer have been addressed in one response. In many cases, we have adjusted or edited our manuscript based on these reviewer comments, outlined in the newly attached manuscript in red text. We have highlighted the modifications to the manuscript by providing page and line information in the below rebuttal.

REFEREE COMMENTS:

Referee #1:

This manuscript appears that it may produce important results but there are some major flaws that make it unclear and difficult to review as follows:

1. The paper should include some description about what the functional assay is measuring and what that means functionally. Right now that is assumed. For example, SRX is not defined anywhere. What are the limitations of this functional measurement? Are there known biases of this approach for each specific fiber type?

Author Response: Thank you for this comment and apologies for not defining and making these points clearer in our original submission. We have addressed this in both the introduction (**Lines 82-88, Page 4**) and results section (**Lines 146-154, Page 6**). We believe this now provides much better context and coherence in our use of the MANT ATP chase assay to study SRX/DRX myosin heads.

2. Figure legends are missing making figures harder to interpret

Author response: Our apologies, this was a mistake during manuscript upload and submission. Figure legends are now presented at the end of the manuscript.

3. More detail on the proteomic sample processing should be given in the manuscript instead of relying on reference to a previous paper.

Author response: We have now amended the relevant section in our methodology ('Myofiber proteomic sample preparation'; **Page 16, line 510**), to provide more information on how we generated proteomic data from single myofiber fragments.

4. Figure 1G and 2G are unclear and there seems likely a mistake in their preparation. The y-axis is the % of which MYH isoform? It seems this figure would require at least two panels showing the % of MYH7 and a second showing the percent of MYH2?

Author Response: Apologies, this was indeed an oversight in figure preparation. The two figures (1G and 2G) have been modified to more easily display the % of myosin isoforms expressed in each myofiber. MYH7, MYH2 and MYH1 are now displayed in different shapes on the plot. For each single myofiber, three separate 'points' (shapes) represent the accumulate relative abundance of MYH 7, 2 and 1.

5. Figure 1M - the points should be colored according to which group they are in to enhance clarity.

Author Response: This has been done.

6. There appears to be no mention of multiple testing correction anywhere, which is required any time multiple tests are performed. Please use corrected p-values for all comparisons. For example, use Benjamini Hochberg for 1H, 1J, 1K, 1L, 1M, 2H, and 2I.

Author Response: Thank you for the comment. We used a corrected p-value method that combines both biological relevance (fold-change) and statistical significance (p-value) to determine a threshold for differential abundance (PMID 22321699). This approach has been used by our, and other, research groups in the field of muscle biology (e.g. PMID 35186464; PMID: 39971958, PMID: 33436631, PMID: 35638262.). We acknowledge that the work performed in this manuscript (as a 'proof-of-principle') is done so with limitation (as acknowledged in our limitations sections) and that further validation of the work will need to be performed. But, by demonstrating the utility and application of SMyoMFO in this work, this can be accomplished.

7. Despite major differences in proteomic depth between control and myopathy, the data appears to have been beautifully normalized (Fig 2H and 2I). The statistical section of the methods is vague with regard to how this was achieved. Please include the code for all data analyses on github or zenodo and include more details in the methods section.

Author response: We would like to refer the reviewer to the Differential abundance section in the methods, where we describe that we followed the *limma* workflow, including a step for quantile normalization. Moreover, as the data has been computationally pooled by a median pseudobulk approach, the differences in depth between the two conditions are reduced for the differential abundance analysis visualized in the volcano plots from figure 2H and 2I. Code used for analysing proteomic data sets has been included on Github, with information included in the manuscript (**Page 19, Line 640-643**) of the location.

8. All the raw mass spectrometry data must be shared publicly and checked before manuscript acceptance. That section of the method currently lacks the data coordinates.

Author Response: These data have been deposited on the ProteomeXchange Consortium via the PRIDE partner repository, under the accession **PXD061287**. These data will be made publicly available upon formal acceptance and publication of the manuscript. Reviewers are able to access these data through the same accession code alongside a reviewer specific account:

Username: reviewer_pxd061287@ebi.ac.uk

Password: cEkWZPDIfGAb

These details have also been updated in the manuscript, **Page 19, Line 640-643**

9. Why were type 1 fibers compared to control only for ACTA1- and why were type 2a only compared to control for TNNT- fibers?

Author Response: We took this approach for a number of reasons. First, anecdotally, ACTA1 and TNNT patients present with enriched type I and type IIa myofibers, respectively. We support this finding with our proteomic subtyping analyses conducted in Figure 2G. It was therefore of interest to ascertain whether these enriched myofiber subtypes presented with similar/different proteomic profiles compared to control. Thus we performed control vs disease Pseudobulk analysis of type I (con ~ ACTA1) and type IIa (con ~ TNNT) in the separate forms of nemaline myopathy. To help disseminate this to the readership, we have added the following section into our results section (**Page 8, Line 243**).

“We therefore aimed to determine whether the elevated prevalence of specific myofiber subtypes observed in ACTA1- and TNNT1-associated SkM exhibits comparable or distinct proteomic profiles relative to their respective healthy control type I and type IIa myofibers.”

10. What is $Xiao < 0.05$?

Author Response: The Xiao (π -value) based approach to determine significance combines fold-change and p-value of the differential target, as described by Xiao et al., 2014 (PMID: 22321699). We report the use of Xiao as a means to detect adjusted significance in our methodology section (**Page 17, from line 553**).

Referee #2:

This manuscript reports the development and use of a two-step approach to describe single myofiber proteomic characterization coupled with a myosin biophysical ATPase measurement. The novel approach is reported for fibers from health humans and is then advanced to two muscle diseases. A rich dataset here and in future studies using this novel approach will be valuable to the scientific community. Additions noted below may introduce possibilities to further advance the field.

Author response:

Dear reviewer. Thank you very much for your evaluation of our work and manuscript and for the positive comments regarding the methodologies impact to the scientific community and of the value of the data set. We have responded to your original comments in the below section (highlighted in red) and made accompanying amends to the manuscript (also in red text).

1) Introduction:

a. Please use patient-first (person-first) language, e.g., rather than "Myopathy patients" write "Patients with myopathy".

Author response:

Terminology/language has been amended throughout the manuscript.

b. There is a lack of information on myosin SRX-DRX states which is needed for readers not familiar with resting myosin ATPases; a couple of sentences on this is warranted earlier than that in the results and discussion.

c. Along with the brief description of myosin SRX, a sentence rationalizing the prioritization of this resting (non-contraction) sarcomeric characterization would be helpful as well, again prior to what is written in the discussion.

Author response: Apologies to both of the above two points (b. and c.), this was an over-sight on our part. We have now included a brief description on both the biology of myosin SRX/DRX state (to address your comment above), and the importance of myosin head state in the context of muscle physiology, tissue and whole-body metabolism and the growing identification of SRX/DRX dysregulation in states of muscle aberrancy. These new additions to the manuscript are found in **line 80 to 91 (page 4)**.

2) Results:

a. Avoid the word "shape" as this implies that protein conformation or an imaging technique was utilized to evaluate myosin.

Author response: Thank you for the suggestion. This has been modified throughout the manuscript.

b. Can any deduction be made in regard to thick vs thin filament proteins, for example, within the results for the nemaline myopathy fiber measurements? Or, in Fig 1 J, thin filament proteins tend to cluster (top blue) apart from thick filament proteins (bottom blue). Is this trend more noticeable than these 5 proteins labelled?

Author response: *(If we have interpreted this point correct)*

This is something we looked into originally, but we observed no specificity for fiber type or thick/filament proteins beyond that demonstrated in Figure 1J. Our analysis of *ACTA1-type1* and *TNNT1-typeIIa* highlighted several isoform specific proteins that were differentially regulated between these sample sets (described on **Page 9, Line 294**). However, these same isoforms were also differentially expressed in control type I and type IIa myofibers. In fact, these isoform specific proteins showed a reduction in delta-change in our nemaline myopathy analysis compared to control, data analysis that formulates Figure 2N and 2O. We have now amended Figure 2O to highlight a large selection of these proteins to support readership.

c. In Fig 1A and elsewhere, it should be made clear that the ATPase measured is during relaxation/ in a resting fiber.

Author response: This is a good point. Throughout the manuscript, we have now stated that all myofibers were in a resting/non-contracting state (e.g. Line 148). Specifically we have addressed this in the methodology section (Line 488, Page 15) and in the legends for figures 1 and 2.

d. Fig legends for Fig 1 and 2 are missing from the composite pdf.

Author response: Our apologies, this was a mistake during manuscript upload and submission. Figure legends are now presented at the end of the manuscript.

3) Discussion:

a. In describing advanced characteristics of 'fast' and 'slow' fibers, it seems that the role of the motor neuron should minimally be noted.

Author response: Thank you for this suggestion. We have now included a section in the introduction noting the involvement of the motor neuron and the motor unit, in the specificity and diversity of myofiber subtypes (Page 4, Line 100-103). This seems to us to be the most appropriate place for this information.

b. Another limitation not mentioned is sex and age discrepancies of patients in the control vs myopathy groups, i.e., controls are all male while 4 of 6 in the myopathy group are female. How might the very large age differences in control vs myopathy groups affect the results?

Author response: We agree with the reviewer that both of these points are indeed very valid limitations to our proof-of-principle experiments. It will be crucial, and very interesting, for future experiments to now use our single myofiber work flow in a much larger, and well balanced (sex and age) cohort of SkM biopsies to validate and further expand upon our initial findings.

We have acknowledged the age and sex limitations of our SkM sample set within the limitations section of our manuscript (Line 401-404, Page 12).

Dear Dr Seaborne,

Re: JP-RP-2025-288363R1 "Integrated single cell functional-proteomic profiling of human skeletal muscle reveals a shift in cellular specificity in nemaline myopathy" by Robert A Seaborne, Roger Moreno-Justicia, Jenni Iaitila, Christopher T.A. Lewis, Lola Savoure, Edmar Zanoteli, Michael Lawlor, Heinz Jungbluth, Atul S Deshmukh, and Julien Ochala

Thank you for submitting your manuscript to The Journal of Physiology. It has been assessed by a Reviewing Editor and by 2 expert referees and we are pleased to tell you that it is acceptable for publication following satisfactory revision.

REVISION CHECKLIST:

We look forward to receiving your revised submission.

Yours sincerely,

Paul Greenhaff
Senior Editor
The Journal of Physiology

REQUIRED ITEMS

- The reference list must be in alphabetical order, rather than numbered, to comply with our Journal format.
- You must upload original, uncropped western blot/gel images (including controls) if they are not included in the manuscript. This is to confirm that no inappropriate, unethical or misleading image manipulation has occurred. These should be uploaded as 'Supporting information for review process only'. Please label/highlight the original gels so that we can clearly see which sections/lanes have been used in the manuscript figures. For more information, see: <https://physoc.onlinelibrary.wiley.com/hub/journal-policies#imagmanip>.

EDITOR COMMENTS

Reviewing Editor:

Your work has been evaluated by the original reviewers. There is a lingering concern with respect to the statistical reporting and the lack of adjusted p values. This is indeed a concern; however, the technical advancement in this paper is significant and will be a good resource to the skeletal muscle field. I suggest that the authors be very clear throughout about the "preliminary" and "proof-of-concept" nature of this work, and be very explicit about the usage of non-adjusted p values in the Abstract, Methods and Results, as well as spend time discussing this in the limitations section. I also suggest that the title be changed to: "Integrated single cell functional-proteomic profiling of human skeletal muscle reveals a potential shift in cellular specificity in nemaline myopathy: A proof-of-concept exploratory study".

Senior Editor:

Thank you for the revised manuscript and rebuttal document which has been considered by the same reviewing editor and reviewers that considered the original submission. There has been a split decision on the merits of the manuscript, with Reviewer 1 believing the inclusion of corrected p-values by the authors is essential to avoid incorrect data interpretation due to poor statistical analysis. The Reviewing Editor believes that on balance, the technical advancement in the manuscript is significant and important and that the paper could be impactful. However, at the moment, the lack of transparency regarding the limitations of the statistical approach used requires the authors to be very up front and transparent if the paper is to proceed further. In an attempt to help the authors, the Reviewing Editor has made a number of suggestions, including labeling this study as "preliminary" or "proof of concept" in the title and the abstract. Furthermore, the authors should be very clear throughout about the "preliminary" and "proof-of-concept" nature of this work and be very explicit about the usage of non-adjusted p values in the Abstract, Methods and Results, as well as spending time discussing this in the limitations section.

REFEREE COMMENTS

Referee #1:

The authors have effectively addressed several points; however, they have not addressed the most critical issue, which fundamentally undermines the validity of their conclusions. Their response regarding the use of uncorrected p-values remains inadequate. While they reference an approach that combines p-values and fold changes into a new score, this does not account for proper multiple testing correction or adjust p-values as they appear on a volcano plot. Without appropriate p-value correction, the reported changes are likely to include random variation rather than true biological signals. The

justification that previous papers have also neglected proper statistical corrections does not validate this approach. Correcting for multiple testing is essential and must be implemented.

A medium-priority concern is that the GitHub repository remains private, preventing a full assessment of the provided materials.

Additionally, a minor issue is that the line numbers referenced in the authors' response do not align with either the marked or clean version of the manuscript. While I was able to locate the relevant sections, greater attention to detail in future submissions would be beneficial.

Referee #2:

The development and use of a novel two-step approach is used to describe single myofiber proteomic characterization coupled with a myosin biophysical ATPase measurement in fibers from healthy humans and humans with neuromuscular diseases. The dataset reported and future studies using this novel approach will be valuable to the scientific community. Limitations and advances to the field are valid and robust in regard to both results and approaches.

END OF COMMENTS

Author response to address editorial and reviewer comments, with regards:

Manuscript:

Integrated single cell functional-proteomic profiling reveals a shift in myofiber specificity in human nemaline myopathy: a proof-of-principle study

EDITORS COMMENTS

Reviewing Editor:

Your work has been evaluated by the original reviewers. There is a lingering concern with respect to the statistical reporting and the lack of adjusted p values. This is indeed a concern; however, the technical advancement in this paper is significant and will be a good resource to the skeletal muscle field. I suggest that the authors be very clear throughout about the "preliminary" and "proof-of-concept" nature of this work, and be very explicit about the usage of non-adjusted p values in the Abstract, Methods and Results, as well as spend time discussing this in the limitations section. I also suggest that the title be changed to: "Integrated single cell functional-proteomic profiling of human skeletal muscle reveals a potential shift in cellular specificity in nemaline myopathy: A proof-of-concept exploratory study".

Senior Editor:

Thank you for the revised manuscript and rebuttal document which has been considered by the same reviewing editor and reviewers that considered the original submission. There has been a split decision on the merits of the manuscript, with Reviewer 1 believing the inclusion of corrected p-values by the authors is essential to avoid incorrect data interpretation due to poor statistical analysis. The Reviewing Editor believes that on balance, the technical advancement in the manuscript is significant and important and that the paper could be impactful. However, at the moment, the lack of transparency regarding the limitations of the statistical approach used requires the authors to be very up front and transparent if the paper is to proceed further. In an attempt to help the authors, the Reviewing Editor has made a number of suggestions including labelling this study as "preliminary" or "proof of concept" in the title and the abstract. Furthermore, the authors should be very clear throughout about the "preliminary" and "proof-of-concept" nature of this work and be very explicit about the usage of non-adjusted p values in the Abstract, Methods and Results, as well as spending time discussing this in the limitations section.

Author Response: To the reviewing and senior editor. Thank you very much for your careful and considered evaluation of our manuscript (v2), the rebuttal and of the reviewers further concerns. We have adjusted the manuscript in order to better reflect the limitations of our analyses. We have amended the title to better reflect the 'proof-of-principle' nature of work, as per the reviewing editors suggestion. However, we have had to further amend the manuscript title to meet The Journals character limit.

There is now a discussion regarding the statistical handling of our experiments with the limitations section as well as regular referral to non-adjusted p-values throughout the abstract, methods and results sections. Further amendments have been made through the abstract, introduction, methods, results and figure legends. All amendments made to the manuscript can be found on the new version, highlighted in red text.

REFEREE COMMENTS:

Referee #1:

The authors have effectively addressed several points; however, they have not addressed the most critical issue, which fundamentally undermines the validity of their conclusions. Their response regarding the use of uncorrected p-values remains inadequate. While they reference an approach that combines p-values and fold changes into a new score, this does not account for proper multiple testing correction or adjust p-values as they appear on a volcano plot. Without appropriate p-value correction, the reported changes are likely to include random variation rather than true biological signals. The justification that previous papers have also neglected proper statistical corrections does not validate this approach. Correcting for multiple testing is essential and must be implemented.

Author Response: We appreciate the reviewers concerns regarding the use of uncorrected/unadjusted p-valued within our manuscript. We also acknowledge that such FDR adjustments is standard practise for investigating biological outcomes in experimental studies. Nonetheless, we would like to clarify that our study primarily stands as a proof-of-principle for the protein function-omic methodology employed on single cells/myofibers, rather than a definitive biological investigation, per se. That is, our main objective of this work is to demonstrate the power of single myofiber protein function-omics, rather than drawing firm biological conclusions itself. In this case, using Xiao to provide a compositive score of significance, balancing p-value and fold-change, enables meaningful and informative patterns within our small sample set to be drawn. We acknowledge that in order to use our single myofiber protein function-omics methodology to make firm biological conclusions both corrected p-values, and larger sample sizes must be used. We see this manuscript as an initial demonstration of what can be done, paving the way for future work to use this approach with more stringent and rigorous statistical framework to help us understanding skeletal muscle biology to a greater extent.

To better reflect these points, we have made changes to our manuscript. Throughout the abstract, methodology and results sections, we have acknowledged that unadjusted, compositive p-values are used to analyse the data. We have also dedicated a large component of our limitations sections to properly addressing the use of unadjusted p values within our experiments, that our work is a proof-of-principle or pilot experiment demonstrating the utility of single myofiber protein function-omics and that work is now needed to be done to validate the biological conclusions of our findings.

The correspondence in the limitations section can be found from page 12, line 422. We have made specific reference to using unadjusted Xiao p-values throughout the methods (page 17, line 593-597), results (page 6 line 184: page 7 line 202, line 224-226: page 8 line 259-260; page 9 line 317-320).

A medium-priority concern is that the GitHub repository remains private, preventing a full assessment of the provided materials.

Author Response: Thank you for this. The GitHub repository should be publicly available.

Additionally, a minor issue is that the line numbers referenced in the authors' response do not align with either the marked or clean version of the manuscript. While I was able to locate the relevant sections, greater attention to detail in future submissions would be beneficial.

Author Response: Our apologies for this oversight.

Referee #2:

The development and use of a novel two-step approach is used to describe single myofiber proteomic characterization coupled with a myosin biophysical ATPase measurement in fibers from healthy humans and humans with neuromuscular diseases. The dataset reported and future studies using this

novel approach will be valuable to the scientific community. Limitations and advances to the field are valid and robust in regard to both results and approaches.

Author Response: We thank the reviewer for the positive comments regarding the methodology and the data, and for their time in carefully reviewing our manuscript.

Dear Dr Seaborne,

Re: JP-RP-2025-288363R2 "Integrated single cell functional-proteomic profiling reveals a shift in myofiber specificity in human nemaline myopathy: a proof-of-principle study" by Robert A Seaborne, Roger Moreno-Justicia, Jenni Iaitila, Christopher T.A. Lewis, Lola Savoure, Edmar Zanoteli, Michael Lawlor, Heinz Jungbluth, Atul S Deshmukh, and Julien Ochala

We are pleased to tell you that your paper has been accepted for publication in The Journal of Physiology.

Yours sincerely,

Paul Greenhaff
Senior Editor
The Journal of Physiology

If you would like to receive our 'Research Roundup', a monthly newsletter highlighting the cutting-edge research published in The Physiological Society's family of journals (The Journal of Physiology, Experimental Physiology, Physiological Reports, The Journal of Nutritional Physiology and The Journal of Precision Medicine: Health and Disease), please click this link, fill in your name and email address and select 'Research Roundup':
<https://www.physoc.org/journals-and-media/membernews>

- You can help your research get the attention it deserves! Check out Wiley's free Promotion Guide for best-practice recommendations for promoting your work at: www.wileyauthors.com/eeo/guide. You can learn more about Wiley Editing Services which offers professional video, design, and writing services to create shareable video abstracts, infographics, conference posters, lay summaries, and research news stories for your research at: www.wileyauthors.com/eeo/promotion.

EDITOR COMMENTS

Reviewing Editor:

Thank you for making the requested changes to the manuscript. Congratulations on a nice publication. A valuable addition to the literature.

Senior Editor:

Thank you for addressing the comments raised. The manuscript is now more balanced and transparent. Congratulations.